# Muscle Synergy during Wrist Movements Based on Non-Negative Tucker Decomposition

**DOI:** 10.3390/s24103225

**Published:** 2024-05-19

**Authors:** Xiaoling Chen, Yange Feng, Qingya Chang, Jinxu Yu, Jie Chen, Ping Xie

**Affiliations:** 1Key Laboratory of Measurement Technology and Instrumentation of Hebei Province, Institute of Electric Engineering, Yanshan University, Qinhuangdao 066004, China; xlchen@ysu.edu.cn (X.C.); fygtxgc@163.com (Y.F.); qingyachang1122@163.com (Q.C.); yujinxu@stumail.ysu.edu.cn (J.Y.); 2Key Laboratory of Intelligent Rehabilitation and Neuromodulation of Hebei Province, Institute of Electric Engineering, Yanshan University, Qinhuangdao 066004, China; 3School of Physical Education, Yanshan University, Qinhuangdao 066004, China; wkb@ysu.edu.cn

**Keywords:** EMG, muscle synergy, matrix factorization, non-negative matrix factorization, non-negative Tucker decomposition

## Abstract

Modular control of the muscle, which is called muscle synergy, simplifies control of the movement by the central nervous system. The purpose of this study was to explore the synergy in both the frequency and movement domains based on the non-negative Tucker decomposition (NTD) method. Surface electromyography (sEMG) data of 8 upper limb muscles in 10 healthy subjects under wrist flexion (WF) and wrist extension (WE) were recorded. NTD was selected for exploring the multi-domain muscle synergy from the sEMG data. The results showed two synergistic flexor pairs, Palmaris longus–Flexor Digitorum Superficialis (PL-FDS) and Extensor Carpi Radialis–Flexor Carpi Radialis (ECR-FCR), in the WF stage. Their spectral components are mainly in the respective bands 0–20 Hz and 25–50 Hz. And the spectral components of two extensor pairs, Extensor Digitorum–Extensor Carpi Ulnar (ED-ECU) and Extensor Carpi Radialis–Brachioradialis (ECR-B), are mainly in the respective bands 0–20 Hz and 7–45 Hz in the WE stage. Additionally, further analysis showed that the Biceps Brachii (BB) muscle was a shared muscle synergy module of the WE and WF stage, while the flexor muscles FCR, PL and FDS were the specific synergy modules of the WF stage, and the extensor muscles ED, ECU, ECR and B were the specific synergy modules of the WE stage. This study showed that NTD is a meaningful method to explore the multi-domain synergistic characteristics of multi-channel sEMG signals. The results can help us to better understand the frequency features of muscle synergy and shared and specific synergies, and expand the study perspective related to motor control in the nervous system.

## 1. Introduction

The central nervous system (CNS) generates different motor outputs through a combination of muscle modular structures that represent muscle synergy [1,2]. In this process, modular structures are often used to organize and coordinate multiple degrees of freedom (DOF) changes among muscles [3]. Previous studies have shown that the study of muscle synergy characteristics is more helpful in understanding the way that CNS controls the production, execution and coordination of human movement [4], which can also be applied in modern myoelectric control for rehabilitation technologies [5,6].

Nowadays, muscle synergy is being studied and has become an important tool to interpret motor control [7]. The existing methods for muscle synergy are largely achieved by applying matrix factorization to surface electromyography (sEMG) data, which are decomposed into some matrices with spatial and temporal patterns [8,9]. Several matrix factorization techniques, such as principal component analysis (PCA) [10,11], independent component analysis (ICA) [12,13] and non-negative matrix factorization (NMF) [14], have been applied to muscle synergy, and notably, the NMF has been the most widely used method [15,16]. In recent years, extensive studies have applied this method to muscle activation during pedal movement [17], walking [18] and elbow movement [19]. These studies mainly focused on muscle synergy in the single-time scale. However, motion control modularization has multi-domain characteristics of space and time, action and rhythmicity, which shows that muscle synergy also has frequency domain characteristics. However, there is a lack of research on muscle synergy multi-domain feature correlation analysis. Therefore, it is of great significance to explore a method that can analyze the multi-domain characteristics of muscle coordination for the study of limb motor control. In addition, research shows that some muscle synergies are task-specific, while others may be shared among different movements and behaviors [20]. In earlier studies, we found that identifying shared muscle synergies required the repeated application of NMF to each exercise or subject, and then relied on indicators such as correlation coefficients to identify shared and task-specific synergies [21]. Therefore, NMF has limitations in the multi-domain characteristics of synergy, and it has become increasingly important to introduce a new method of multi-domain muscle synergy characteristics extraction.

Matrix factorization is the factorization of a matrix into a lower-dimensional matrix, and tensor factorization is the splitting of high-dimensional data into its lower-dimensional components. The core idea of tensor decomposition and matrix decomposition is the same, that is, to split high-dimensional data into low-wiki components. Tensor decomposition is a generalization of matrix decomposition, and it can deal with high-dimensional data, and can capture more complex data relationships and structures. Tensor decomposition can be seen as the multidimensional generalization of matrix decomposition, that is, the extension of a two-dimensional matrix into a low-dimensional matrix to a three-dimensional tensor into a low-dimensional component. Compared to NMF, the tensor decomposition method decomposes the data tensor into a non-negative multidimensional feature matrix that can better explain muscle activation. It can be regarded as a multidimensional expansion form based on matrix data processing [22] and can mine the interaction between multiple dimensions of signals [23]. Since the 1980s [24], tensor decomposition, as an important tool for signal processing, has drawn a lot of important attention [25]. H Yang et al. applied tensor decomposition to image processing and reconstructed images [26]. Ahmed Ebied et al. proposed a synergy effect-based proportional myoelectric control on tensor decompositions [27]. However, the multi-domain synergistic characteristics of wrist flexion and extension of the upper limb based on tensor decompositions have not yet been studied. To further explore the neural control and motor coordination mechanism under different movements of the upper limbs, and explore the multi-domain muscle synergistic characteristics of the multi-channel surface muscle signals, this study introduces the non-negative Tucker decomposition (NTD) method to study the multi-domain muscle synergy under the wrist flexion and extension action of the human upper limbs. In the process of research, we assume that each task component has a task action-specific synergy and that there is a shared synergy between the two task components (the WF and WE motion maintenance phase).

To explore the multi-domain characteristics of synergy muscles, we applied the NMF and NTD methods to analyze the sEMG signal recorded from eight muscles involved in wrist flexion (WF) and wrist extension (WE) movements of 10 healthy people. Firstly, we used NTD to estimate muscle synergies to obtain the time and frequency domain information of the muscle synergy modules. The results will be compared with NMF. And then, NTD was used to extract specific and shared synergies between two movements in a direct way that is less complex than NMF [28]. In this study, we extracted multi-domain muscle synergies by using NTD and showed the superiority of this method in the extraction of muscle synergy. This can help us further understand the motion control mechanism of the nervous system. Our findings on muscle synergy will be of great significance to motor control and even to clinical assessment techniques.

## 2. Material and Methods

### 2.1. Subjects

Ten healthy subjects (7 males and 3 females, right-handed, age range 22–28 years) without any medical and physiological history were selected. Before the experiments, all subjects filled out the Oldfield questionnaire and signed an informed consent. The experiment was performed based on the Declaration of Helsinki and confirmed by the Ethics Committee. All subjects had no strenuous exercise experience recently, and no similar experimental experiences.

### 2.2. Experiments and Data Recording

#### 2.2.1. Experiments

During the experiment, the subject put their right upper limb on a bracket to maintain the forearm in a horizontal position, and the elbow joint bent to 90 degrees. The upper limb bracket had no auxiliary or resistance for the wrist task. All subjects performed wrist movements according to a target picture displayed on a computer screen. Figure 1a shows the schematic diagram for one task. In this experiment, each subject was subjected to perform 10 trials. As Figure 1b shows, each trial mainly involved a relaxed stage for 2 s, a relaxed transition to the WF stage for 2 s, a steady WF stage for 2 s, then a WF transition to the WE stage for 2 s, a steady-stage WE movement for 2 s and another relaxed state for 1 s. All subjects had a rest for 60 s between each session to avoid fatigue.

#### 2.2.2. Data Recording and Preprocessing

According to previous studies, EMG data were recorded with a Trigno wireless EMG system (Delsys Inc., Natick, MA, USA) from 8 muscles of the right upper limb including the biceps brachii (BB), brachioradialis (B), flexor carpi radialis (FCR), palmaris longus (PL), extensor carpi radialis (ECR), extensor digitorum (ED), extensor carpi ulnaris (ECU), and flexor digitorum superficialis (FDS) during the wrist flexion and extension task [29,30]. These muscles were considered the main muscle groups in wrist movements.

Before the experiment, the skin was wiped with 75% medical alcohol to minimize impedance. Before placing the electrodes, we used the alcohol to evaporate to dry the skin. According to anatomical knowledge, and through several specific movements, we found the corresponding muscle position [31] and stuck the electrode along the direction of muscle fibers to the middle position of the corresponding muscle [32]. Before further analysis, the raw signals need to be preprocessed. First, the data segments with severe hand shaking or delayed response were deleted. Then, all signals were downsampled to 500 Hz. Finally, the signals were filtered with a 150 Hz low-pass filter and full-wave rectified to obtain the EMG envelope for further analysis.

### 2.3. Method

Non-negative Tucker decomposition (NTD) was selected for muscle synergy, which is a tensor decomposition model. To evaluate the tensor-based approach for muscle synergy, we introduce NMF as a state-of-the-art benchmark for comparison and discussion.

#### 2.3.1. Non-Negative Matrix Factorization

Here, we used NMF to decompose the pre-processed EMG signals matrix V (channel×time) to obtain two non-negative matrices, namely, synergy matrix W and weighting functions matrix H [14], which were used to analyze the spatial–temporal domain characteristics. Among them, the muscle synergy matrix W reflects the relative weighting of muscles in every module, and the weighting functions matrix H was obtained. The NMF algorithm is as follows:(1)$$V_{ij}\approx {\left(WH\right)}_{ij}=\sum_{\mu -1}^{n}W_{iu}H_{uj}=V_{ij}^{\prime }$$
where n is the number of muscle synergy, i represents the number of the signal, j denotes the number of sampling points, and $V^{\prime }$ represents the reconstructed sEMG signals. To exhibit the weight value of the active muscle based on the muscle synergy vector matrix *W*, the *W* value was normalized in the range from 0 to 1. The muscles were considered synergistic muscles if the weight values were higher than 0.5.

For the determination of the number of muscle synergies, Ranaldi S et al. [33] proposed an Akaike Information Criterion (AIC)-based method for model order selection when extracting muscle synergies via the original Gaussian Non-Negative Matrix Factorization algorithm, and Soomro MH et al. [34] assessed the performance of different initializations of matrix factorization algorithms for an accurate identification of muscle synergies. In this paper, the number of synergy modules k depends on the variability accounted for (VAF) considered before applying NMF [35]. The definition of VAF is as follows:(2)$$VAF=1-\frac{RSS}{TSS}=1-\frac{\sum {\left(V-V^{\prime }\right)}^{2}}{\sum {\left(V\right)}^{2}}$$
where RSS is the sum of residual squares and TSS is the sum of squares. Clark D J et al. [36] found that when the number of synergy modules was small, the decomposition model only covers part of the useful information of the original signal. The threshold is 90%. It is considered that when the VAF value is greater than 90% and the increase in VAF is less than 2% with the increase in the value, then stop the calculation, and select the current k value as the number of muscle synergies.

According to the above analysis, NMF can obtain the characteristics of muscle synergy in time–space domain.

#### 2.3.2. Non-Negative Tucker Decomposition

Tensor decomposition is an effective method to extract high-dimensional data features. This method can be used to extract the hidden potential features of different dimensions in the data structure and the interaction information between different dimensions. The Tucker model is one of the most prominent and commonly used models in tensor decomposition. Figure 2 shows the third-order Tucker tensor decomposition model. Here, the tensor X can be decomposed into a core tensor G and factor matrices W, H and P, which are on different dimensions.

The following takes the tensor of order 3 X as the original tensor as an example. The tensor X is sliced horizontally when the tensor is decomposed; in general, tensors $X\in R^{I_{1}\times I_{2}\times I_{3}}$ of order 3 can be decomposed according to the Tucker model as follows:(3)$$X\approx G\times_{1}B^{\left(1\right)}\times_{2}B^{\left(2\right)}\times_{3}B^{\left(3\right)}$$
where $G\in R^{j_{1}\times j_{2}\times j_{3}}$ is the core tensor, $B^{\left(N\right)}\in R^{j_{n}\times j_{n}}$ is the component matrix spanning each mode, and × is the modulo-n matrix product operator of the tensor. The Tucker decomposition is typically solved using the least squares algorithm, where the factor matrix $B_{k}^{\left(i+1\right)},\ldots ,B_{k}^{3}$ obtained at time K and the updated factor matrix $B_{k}^{\left(1\right)},\ldots ,B_{k+1}^{\left(i-1\right)}$ at time k + 1 are utilized to solve for the least squares solution of the factor matrix:(4)$${\overset{\wedge }{B}}_{k+1}^{\left(i\right)}=\underset{B^{\left(i\right)}}{\arg\min}∥X-f(B_{k+1}^{\left(i\right)},\ldots ,B_{k+1}^{\left(i-1\right)},B_{k+1}^{\left(i+1\right)},\ldots ,B_{k+1}^{\left(3\right)}){∥}_{2}^{2}$$
where i=1,2,3. During each iteration, the least squares algorithm is alternately applied to each factor matrix in order to satisfy the convergence criteria of the algorithm.

Because the time, frequency and channel of sEMG signals are non-negative [37], the factor matrix obtained by NTD decomposition is non-negative. Therefore, it is necessary to apply non-negative constraints on the core tensor G and component matrix B in the calculation process. In addition, the non-negative Tucker model is flexible in determining the number of components in each dimension. Therefore, NTD is well suited for capturing muscle synergy in a concise and interpretable manner.

The difference between tensor decomposition and matrix decomposition lies in how to prepare the original data model for analysis. Therefore, before the multi-domain feature analysis of multi-channel EMG data, the EMG data should be first converted into multidimensional information and a high-order tensor model should be established. Then, the high-order tensor model should be decomposed based on non-negative tensor decomposition to obtain the feature matrices of different domains. Therefore, to analyze the multi-channel EMG data in time, space and frequency domains and other features, we need to transform EMG data into the corresponding multidimensional data information and build a high-order tensor model.

Firstly, conversion of EMG signals to time–frequency domain can enhance the dimension. Wavelet transform can decompose signal in time–frequency domain and extract low-frequency information effectively [38,39]. The complex Morlet wavelet CMOR6-1 with the bandwidth parameter $F_{b}=6\;\mathrm{Hz}$ is used to convert all EMG signals into the time–frequency domain, and the wavelet center frequency $F_{c}=1\;\mathrm{Hz}$. Since the action state has a significant correlation with the energy of EMG signals, the modulus of wavelet coefficient is chosen as the representation of time–frequency characteristics. The data obtained from each experiment will transform the multi-channel EMG into a third-order tensor in the time–space–frequency domain, in which each piece of the tensor is the wavelet transform of the corresponding channel.

After obtaining the third-order tensor model, we choose NTD to decompose the third-order muscle synergy tensor model $X\in R^{i\times j\times n}$. The NTD algorithm is as follows:(5)$$X_{ijn}\approx G\times_{1}W^{i\mu_{1}}\times_{2}H^{j\mu_{2}}\times_{3}P^{n\mu_{3}}$$
where $G\in R^{\mu_{1}\times \mu_{2}\times \mu_{3}}$ is the core tensor, $W^{i\mu_{1}}$, as synergy matrix, reflects the participation degree of each muscle in a certain movement, $H^{j\mu_{2}}$ is weighting functions matrix and reflects the activation degree of the synergistic module and $P^{n\mu_{3}}$ is spectrum component matrix and reflects the spectral information of the synergistic module. Where × is the modulo-n matrix product operator of the tensor. The determination of synergistic muscle was the same as that of NMF. If the P value of the synergy module is higher than 0.5, it is regarded as the main spectral component of the synergy module.

In this study, a method based on NTD is introduced to extract muscle synergies, and the concepts of specific synergies and shared synergies were combined. It was found that there are shared synergies and specific synergies between different movements. We assumed that the tasks in 2 components (WF and WE) have a shared synergy. Constraints are imposed on this [1.5*2 components, 2 components, 2 components] Tucker model. The number of muscle synergies (the spatial pattern components) will be the sum of the specific synergies for each movement and the number of shared synergies between them. Both the core tensor and the repeated pattern are initialized and fixed to identify the spatial pattern components. Task-specific synergies link to a component (movement) pattern, while shared synergies link to the two components. In the alternating least squares iteration stage, due to the physical properties of synergies, non-negative constraints are imposed on the time domain and space domain. After iterative optimization, the component matrix on the multi-dimension is obtained [21].

Before NTD calculation, we firstly need to choose the size of the core tensor. Therefore, we introduce the fitting value (FIT) method to decide the R. The definition of FIT is
(6)$$normresidual=\sqrt{({\left\|Y\right\|}^{2}-{\left\|M\right\|}^{2}}$$
(7)$$FIT=1-\frac{normresidual}{\left\|Y\right\|}$$
where $Y\in R^{I_{1}\times I_{2}\times \ldots \times I_{N}}(N\geq 3)$ is the original tensor; $M\in R^{I_{1}\times I_{2}\times \ldots \times I_{N}}$ is a reconstructed data tensor decomposed by NTD. R is artificially selected, and the range of selection is $\left(1\leq R\leq I_{1})\cap (1\leq R\leq I_{2})\cap (1\leq R\leq I_{3}\right)$. The threshold is 90%. It is considered that when the FIT value is greater than 90% and the increase in FIT is less than 2% with the increase in the value, then stop the calculation, and select the current value as the number of muscle synergies.

According to the above analysis, we used NTD to obtain the multidimensional characteristics of muscle synergy. It can not only decompose the frequency domain characteristics of muscle synergy, but also identify the shared muscle synergy and specific synergy between different movements.

#### 2.3.3. Correlation Analysis of Muscle Synergy

Similar synergies can reflect common functions of movement, while specific synergies can represent differences in tasks. Different individuals may have different muscle synergies in different tasks.

A high correlation suggests that the central nervous system co-activates the muscles in the same group. The similarity between two muscle synergies matrices $W=\left[w_{1},w_{2},\ldots ,w_{n_{1}}\right]$ and $W^{*}=\left[w_{1}^{*},w_{2}^{*},\ldots ,w_{n_{2}}^{*}\right]$ ($n_{1}$ and $n_{2}$ represent the number of muscle synergies, $n_{1}<n_{2}$) was estimated by Pearson’s correlation coefficients r ranging from 0 to 1, and a large value means high similarity. When r>0.5, they are considered similar, as shown in the formula:(8)$$r_{w}(W,W^{*})=\frac{1}{n_{1}}\sum_{k=1}^{n_{1}}\max\left[r(w_{k},w_{l}^{*}){|}_{l=1}^{n_{2}}\right]$$

## 3. Result

### 3.1. Choosing the Optimal Number of Synergies under NTD

In this study, we took the third-order tensor decomposition (time–space–frequency) under different actions as an example to determine the R value of NTD decomposition as determined by FIT. Figure 3 shows the FIT of 10 subjects in different Rs under WF and WE, respectively. With the increase in the number of synergies, the FIT value gradually increases and the rate of increase also gradually reduces. When the number of synergies reaches three, the average FIT value reaches more than 90%, and when the number of synergies increases to four, the average increase in the FIT value is less than 2%. Therefore, in this study, we set the number as three to determine the synergy structure.

### 3.2. Muscle Synergy Extraction

#### 3.2.1. The Muscle Synergy Estimated by NMF

In this study, in order to compare and analyze the differences between NMF and NTD in a muscle synergy analysis, we first decomposed the average EMG signals from 10 trials of the same randomly selected subjects based on NMF decomposition. According to Formula (2), we selected the number of synergistic modules as three to meet the experimental requirements.

As Figure 4a shows, there are six pairs of synergy muscles in module W^(1)^: (FCR-PL, FCR-FDS, FCR-BB, PL-BB, PL-FDS and FDS-BB) and a pair of synergy muscles in module W^(2)^: (ECR-BB) in the WF stage. Similarly, in the WE stage, Figure 4b shows three pair of synergy muscles in module W^(1)^: (ECR-FDS, ECR-BB and FDS-BB) and six pairs of synergy muscles in module W^(3)^: (ED-ECU, ED-ECR, ED-B, ECU-ECR, ECU-B and ECR-B). The CNS activates these synergistic muscle pairs to support the WF and WE stages.

#### 3.2.2. The Muscle Synergy Estimated by NTD

To further explore the space–frequency–time muscle synergy feature in different movement stages, NTD was used to decompose the same sEMG signals as the NMF decomposition. Figure 5 shows that each movement can be decomposed into muscle synergy modules, muscle spectrum components and weighting functions. In Figure 5a, Synergy 1 consisted mainly of the activation of PL and FDS, which were activated in the first 1.5 s of the WF stage. The spectral curve P^(1)^ can be observed, showing that the muscle synergy matrix had a significant activation occur at approximately 13 Hz, with spectrum components mainly for 0–20 Hz; Synergy 2 mainly recruited BB and was activated in the second 1 s of WF stage. It can be observed from the spectral curve P^(2)^ that the muscle synergy matrix had a significant activation occur at approximately 25 Hz, with the spectrum component of the muscle synergy matrix being mainly in the band 10–30 Hz; Synergy 3 mainly consisted of ECR and FCR, which were activated during the first 1 s. The spectral component of the muscle synergy matrix is mainly in the band 25–50 Hz.

In Figure 5b, a significant activation in the weightings of ED and ECU in Synergy 1 was observed in the first second of the WE stage. The muscle synergy matrix has an obvious peak at the frequency of 13 Hz, with the spectral component being mainly in the band 0–20 Hz; Synergy 2 mainly recruited BB and was activated in the second 1 s of WE, with the spectral component being mainly in the band 15–30 Hz, and had obvious peak value at 25 Hz; Synergy 3 consisted primarily of activation of the ECR and B, and was activated between 0.5 and 1.5 s. The spectral component of this muscle synergy matrix is mainly in the band 7–45 Hz.

### 3.3. Muscle Synergy Similarity between Tasks and Subjects

We used NTD to analyze and summarize the synergy results for all 10 healthy subjects in Table 1 and Table 2. All the subjects (10 subjects) could extract three synergy matrices under different movements. To better compare and observe the synergistic similarity of the muscles, in this paper, the muscle synergy similarity of the subjects under the same task and the similarity between different tasks of the same subjects were compared and analyzed.

Calculating the muscle synergy similarity between the same subject and different tasks of the same subject according to Equation (8) found that the muscle synergy among different tasks of the same subject presented moderate similarity (0.57 ± 0.19), and the muscle synergy extracted by different subjects under the same task showed high similarity (WF: 0.693 ± 0.20, WE: 0.782 ± 0.15).

The results showed that for upper limb muscle synergy, although there were differences in muscle synergy among different subjects, there was a certain regularity on the macro level. Similar results were also shown with NMF. In addition, there was a shared synergy between muscle synergies in different movements of WF and WE, which reflects the common function between different movements, while specific synergies determine the formation of different movements.

### 3.4. Shared Synergy and Specific Synergy

In order to compare the extraction performance and results of shared synergy and special synergy through non-negative tensor decomposition, we initially employed the conventional method of extracting shared synergy and special synergy based on non-negative matrix factorization. According to the NMF, the muscle synergy module matrix of WF and WE was obtained and its similarity was analyzed. To obtain the connection between the muscle synergy modules between different movements, the correlation coefficient of each muscle synergy module between WE and WF is shown in Table 3. Spearman correlation analysis showed that WF-W^(3)^ and WE-W^(2)^ muscle synergy modules had high similarity, while other synergistic modules had low similarity, notably, WF-W^(1)^ and WE-W^(3)^ muscle synergy modules had the lowest similarity. On the one hand, it showed that BB, as a shared muscle synergy module, plays a role in both WF and WE movements, and we believe that BB plays a role in fixing the upper arm under WE and WF movements. On the other hand, it suggested that WF-W^(1)^ was the specific synergy of WF, while WE-W^(3)^ was the specific synergy of the WE. Therefore, the results showed that there are both shared muscle synergy modules and specific muscle synergy modules among different actions.

Then, we constructed a tensor model based on time–motion–space, and decomposed the model using NTD to explore the special and shared synergistic muscles under WF and WE. The tensor of time–movement–space is decomposed by NTD, and the factor matrix of each dimension is obtained to show the meaning of each component. As Figure 6 shows, the movement component corresponding to the synergy module (W1) has a larger amplitude in the WE, so this synergy module is considered to be WE-specific synergy, which mainly reflects the contribution of ED, ECU, ECR and B. The same is true for W2, so the muscle synergy module (W2) is the specific muscle synergy module of WF, which mainly reflects the contribution of FCR, PL and FDS. And muscle synergy module (W3) is the shared muscle synergy module of WF and WE, which mainly reflects the contribution of BB in the whole process. The results are consistent with the results obtained by NMF.

## 4. Discussion

The significant contribution of this paper is to explore the multi-domain muscle synergy during wrist movements based on the NTD method. The results presented two synergistic flexor pairs (PL-FDS and ECR-FCR) in the WF stage, these muscle synergy matrix spectrum components are mainly in the respective bands 0–20 Hz and 25–50 Hz, and two synergistic extensor pairs (ED-ECU and ECR-B) in the WE stage, with these muscle synergy matrix spectrum components being mainly in the respective bands 0–20 Hz and 7–45 Hz. The activation degree of BB in the WE and WF stages is higher, and the spectral component and activation time are highly coincident. We also found that the flexor muscles FCR, PL and FDS were the specific synergy modules of WF. The extensor muscles ED, ECU, ECR and B were the specific synergy modules of WE, and BB is a shared muscle synergy module of WE and WF, which plays a role of stabilizing the upper limb.

### 4.1. Space–Frequency–Time Characteristics of Muscle Synergy

For the feature analysis of muscle coordination, previous studies mainly extracted and analyzed the spatio-temporal features of muscle coordination based on non-negative matrix decomposition. Xiao L et al. introduced the non-negative matrix factorization (NMF) method to explore the muscle activation patterns and synergy structure under six types of movements [40]. De Marchis C et al. analyzed the distribution of neural actuations among muscles that work together during the execution of a free-pedal task based on non-negative matrix factorization [17]. However, these studies neglected the frequency domain features of muscle coordination. Previous studies have shown that the motion control module has temporal and rhythmic characteristics. Xie P et al. proposed a novel method, named time–frequency non-negative matrix factorization (TF-NMF), to explore the time-varying regularity of muscle synergy characteristics of multi-channel surface electromyogram (sEMG) signals at different frequency bands [41]. However, these studies show that the research based on NMF cannot analyze the multi-domain muscle synergy features. In this study, the potential application of a higher-order tensor model in extracting multi-domain synergies was explored. We found that compared to NMF, the NTD method could extract multi-domain features of muscle synergy. The results showed two synergistic flexor pairs (PL-FDS and ECR-FCR) in the WF stage, with matrix spectrum components mainly for 0–20 Hz and 25–50 Hz. Two synergistic extensor pairs (ED-ECU and ECR-B) in the WE stage, with muscle synergy matrix spectrum components mainly for 0–20 Hz and 7–45 Hz. The activation degree of BB in the WE and WF stages was higher, and the spectral component and activation time were highly coincident. The results indicated that the synergy of flexor muscles mainly existed in the 0–50 Hz band during the WF stage, and the synergy of extensor muscles mainly existed in the 0–45 Hz band during the WE stage. In conclusion, because oscillations occur synchronously in different parts of the motor cortex [42], in the neuromuscular system, motor neurons transmit control commands to the innervated muscle fibers through action potentials, causing muscle contraction. When a control command is transmitted to all muscle fibers of the flexor and extensor muscles at the same time, it will contract synchronously. This phenomenon is called muscle co-activation, also known as synergistic contraction [43]. The possible explanation is that during the isometric compensation of low horizontal forces, the synchronous oscillations of the cortical muscles are mostly reflected in the beta (15–35 Hz) and gamma (35–50 Hz) bands [38], and the higher amplitude of frequencies occurs in the lower range of frequencies. This is concurrent with studies suggesting that lower frequencies correspond to motoneuronal drives relevant to the task [44], which is similar to the conclusion that the movement control information transmitted by neurons was primarily in the beta frequency band [45]. This result may reflect functional linkages between motor cortex neurons and motoneurons supplying the hand muscles. The co-activation time of the synergy muscle pairs was different during the maintenance phase of exercise. The time-varying coefficient can be used to observe the changes in the activation level of muscle-tendon units (MTUs). The possible explanation is that the excitation contribution of each muscle synergy pair was different. And the functional activation of muscles at specific frequencies at specific times constitutes the state of motion. The activation degree of BB in WE and WF was higher. And the spectral components were highly coincident. Their activation time was in the second half of the movement. This suggests that the BB muscle plays a support role in upper limb movement, with the possible explanation being that energy metabolism and the number of activated cells in the muscle will gradually change and cause fatigue during the period of movement maintenance. To overcome fatigue and maintain a steady state, the CNS increases muscle activity, which maintains the stability of movement.

We concluded that the CNS achieves body movement by controlling the coordination of the related muscles. The synergistic muscle pairs are controlled as a whole group. Therefore, the response of synergistic muscle pairs was realized through co-activation inputs [46]. And the muscle synergy in healthy adults showed significant activation before and after the phases of a task [47].

### 4.2. Shared and Specific Synergy

This study analyzed the shared and specific characteristics of muscle synergy under different movement stages. The results show that FCR, PL and FDS are activated during the WF stage and exert a force effect. In the WE stage, ED, ECU, ECR and B are activated to exert force in the whole WE movement. Ahmed Ebied et al. showed that non-negative tensor decomposition can be used for the direct extraction of shared muscle synergies and special muscle synergies by exploring the potential application of higher-order tensor models in myoelectric control [27]. According to the analysis, the ED, ECR, ECU and B muscles are specific synergy modules of WE, while the FCR, PL and FDS are specific synergy modules of WF. BB was a shared-muscle synergy module, which supports wrist flexion and extension and maintains the stability of the upper limb [18]; this conclusion is consistent with the results of studies on muscle activation and muscle synergism in different movement modes of the upper limb [40]. It was found that the action potentials used to produce various movements were different. And the muscle synergy under different movements has shared synergy modules and specific synergy modules, and these modules cooperate and coordinate to complete the movement.

From the analysis results of muscle synergy, different task targets may invoke shared muscle synergy, and the specificity of task targets is reflected in their specific muscle synergy. Specific synergy modules of different movements work together with shared synergy modules to produce flexible and diverse movement patterns. These results indicate that the CNS reprograms new synergies after task objectives change [48]. Muscle synergy can effectively solve the complex multi-degree-of-freedom motion control problem, and can produce different movements with fewer variables. To better understand the neural mechanisms of motor control, muscle synergy was compared by NMF and NTD analysis. NMF can extract the muscle synergy of WF and WE separately, and obtain the muscle synergy between WF and WE through correlation analysis. However, NTD can directly decompose and extract shared and specific synergistic modules between different movements, which has the potential to directly identify synergies, as shown in Figure 6. NTD provides a way to decompose and identify the muscle synergy between movements by adding movement information to the tensor. It is more convenient and effective than traditional matrix analysis algorithms. NTD can identify shared and specific synergies across movements.

This study utilized wavelet decomposition and non-negative tensor decomposition to investigate the time–frequency–space characteristics of muscle coordination. The limitation of this study is that the exploration frequency band ranged from 0 to 150 Hz, without further subdivision into frequency bands such as alpha, beta and gamma bands. In future studies, we will analyze multi-domain cooperative features based on specific frequency bands. In addition, we only explored the multi-domain muscle synergy features in healthy people and did not consider patients with motor dysfunctions, such as stroke. For patients, the activation and inhibition of multi-domain muscle synergies may be relatively diminished, resulting in a joint response to motor control. Therefore, exploring multi-domain muscle synergies provides a more theoretical basis for clinical evaluation applications.

## 5. Conclusions

This study provided a proof of concept for the application of NTD in the high-order tensor muscle synergy. Compared with traditional NMF, the NTD method, as the most currently used matrix factorization model for synergy, can also obtain the spectral components of muscle synergy. It provides more comprehensive information on muscle synergy for the understanding of motor mechanisms. It can also directly obtain the specific synergy and shared synergy of different movements. The results indicate that tensor decomposition is more effective than matrix decomposition in extracting the hidden data and multi-domain synergistic features from sEMG data. Therefore, these results indicate the potential application value of NTD as a muscle synergy analysis method. Our findings on muscle synergy will be of great significance to motor control and even to clinical assessment techniques.

## Figures and Tables

**Figure 1 sensors-24-03225-f001:**
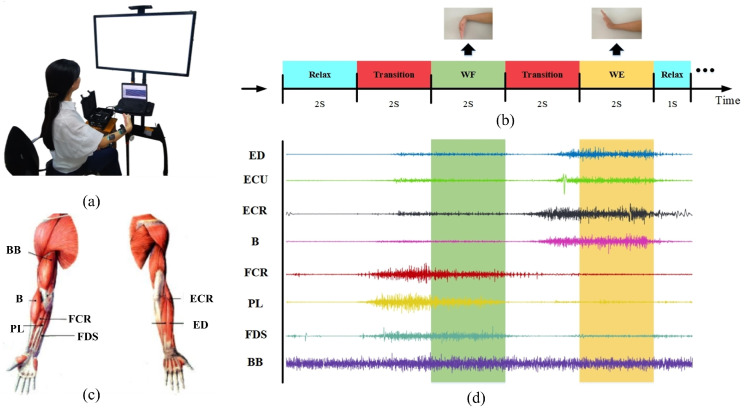
Experimental setup: (**a**) The experiment process scene. (**b**) The flow chart of experimental tasks. (**c**) The diagram of the electrode position. (**d**) The EMG signals were acquired.

**Figure 2 sensors-24-03225-f002:**
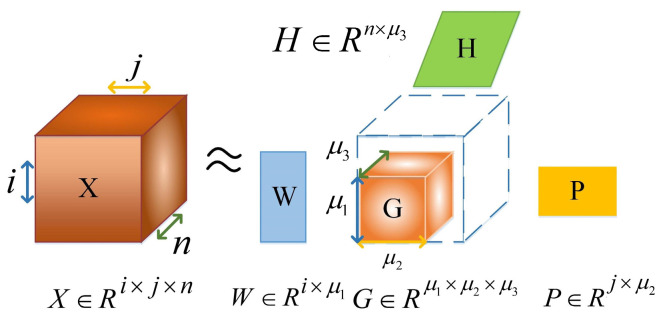
Third-order Tucker tensor decomposition model.

**Figure 3 sensors-24-03225-f003:**
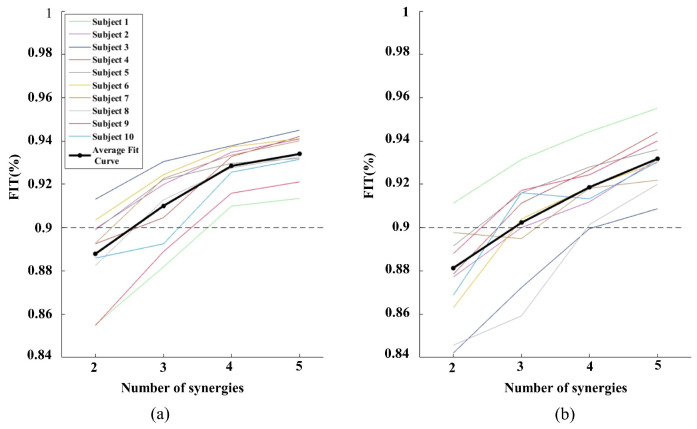
FIT of 10 subjects in different Rs under WF and WE: (**a**) FIT of different Rs under WF (**b**) FIT of different Rs under WE. The dashed line indicates the threshold > 90%.

**Figure 4 sensors-24-03225-f004:**
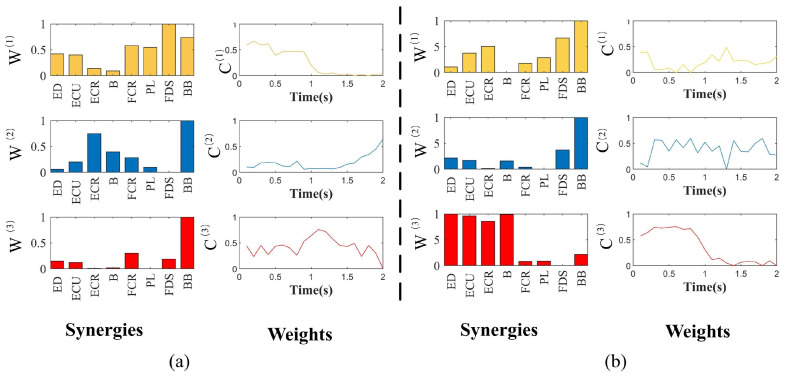
The synergy matrices and weighting functions estimated by NMF for WF and WE: (**a**) The synergy muscle analysis in WF stage by NMF. (**b**) The synergy muscle analysis in WE stage by NMF. Each bar W^(1)^ represents the relative spatial information of muscle co-activation within each synergy. Weighting coefficients were normalized by maximum values under the synergy (1st column). Each waveform C^(1)^ represents the temporal activation pattern of the synergy related to individual muscle-weighting components (2nd column). Synergy 1: W^(1)^, C^(1)^; Synergy 2: W^(2)^, C^(2)^; Synergy 3: W^(3)^, C^(3)^.

**Figure 5 sensors-24-03225-f005:**
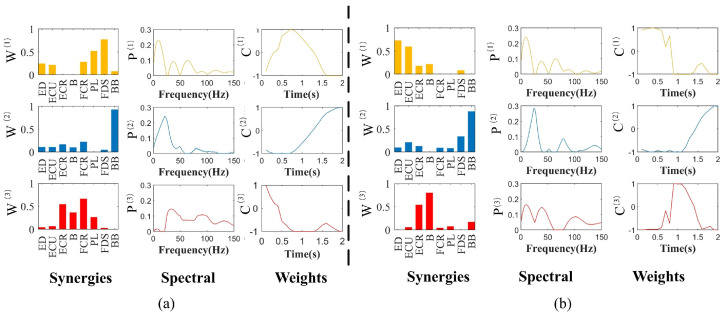
The synergy matrices, spectral components and weighting functions estimated by NTD for WF and WE: (**a**) The synergy muscle analysis in WF stage by NTD. (**b**) The synergy muscle analysis in WE stage by NTD. Each bar W^(i)^ represents the relative spatial information of muscle coactivation within each synergy. Weighting coefficients were normalized by maximum values under the synergy (1st column). Each waveform P^(i)^ represents the muscle spectral components of the synergy module (2nd column). Each waveform C^(i)^ represents the temporal activation pattern of the synergy related to individual muscle-weighting components (3rd column). Synergy 1: W^(1)^, P^(1)^, C^(1)^; Synergy 2: W^(2)^, P^(2)^, C^(2)^; Synergy 3: W^(3)^, P^(3)^, C^(3)^.

**Figure 6 sensors-24-03225-f006:**
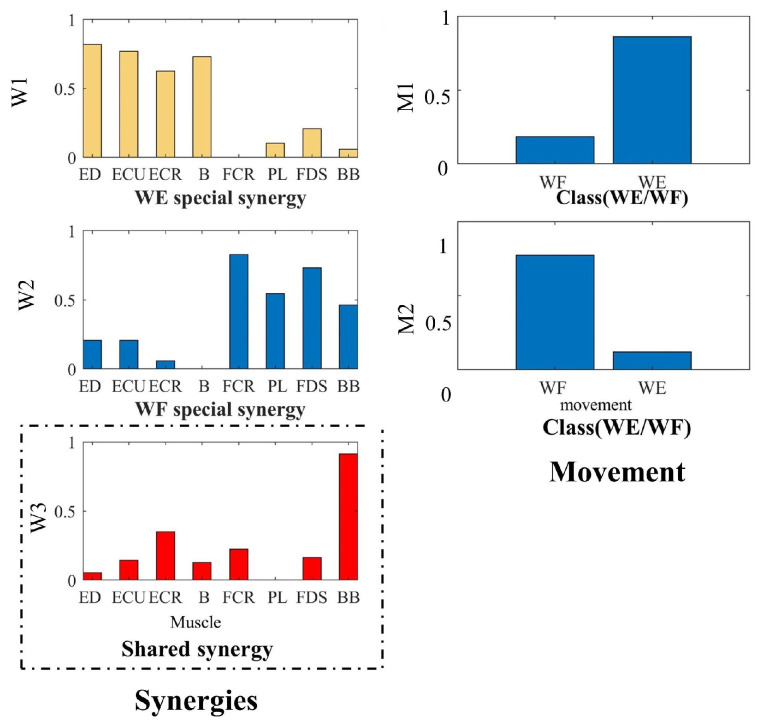
NTD for shared synergy and specific synergy. Each bar W(i) represents the relative spatial information of muscle co-activation within each synergy. Weighting coefficients were normalized by maximum values under the synergy (1st column). Each bar M(i) represents the movement (spatial) component of WF and WE, which is described by the topographic map of amplitude. Weighting coefficients were normalized by maximum values under the synergy (2nd column). W1: WE specific Synergy; W2: WF specific Synergy; W3: Synergy surrounded by a dashed line is a task-shared synergy.

**Table 1 sensors-24-03225-t001:** The synergistic muscles of all subjects estimated by NTD during WF stage.

Synergy Module Subjects	W^(1)^	W^(2)^	W^(3)^
S1	PL FDS BB	ECR FCR	FCR BB
S2	BB	FCR FDS	ECRFDS BB
S3	FDS BB	ECR FCRPL	FCR BB
S4	PL FDS	FCR PL	ECU ECR B
S5	FDS	ECR B	FCR PL BB
S6	PL	BB	ECR FCRPL
S7	FCR PL	PL FDS	ECR
S8	ED ECU FDS	B FCR PL	BB
S9	FCR PL	FDS BB	PL BB
S10	FDS BB	ED FCR	FCR PLFDS

**Table 2 sensors-24-03225-t002:** The synergistic muscles of all subjects estimated by NTD during WE stage.

Synergy Module Subjects	W^(1)^	W^(2)^	W^(3)^
S1	BB	ECR PL	ED ECR B
S2	ECU ECR B	BB	ED ECUECR
S3	ECU	BB	FDS PL
S4	ED ECU ECRB	ECU	PL BB
S5	ED ECR B	BB	B FDS
S6	ECU ECR B	BB	ECU B FDS
S7	ECR B	ED BB	ECR B
S8	ED B	ECR BB	BB
S9	ECR B	BB	ECU ECR B
S10	ECR B	BB	ECR B

**Table 3 sensors-24-03225-t003:** Correlation coefficient of each muscle synergy module between WF and WE.

Muscle Synergy Module	WE-W^(1)^	WE-W^(2)^	WE-W^(3)^
WF-W^(1)^	0.571	0.476	−0.762
WF-W^(2)^	0.191	−0.071	0.167
WF-W^(3)^	0.333	0.7619	−0.286

## Data Availability

All data that support the findings of this study are available on request from the corresponding author.

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
