# Peer review of "Muscle Synergy during Wrist Movements Based on Non-Negative Tucker Decomposition"

_sensors, 2024, doi:10.3390/s24103225_

Round 1
Reviewer 1 Report
Comments and Suggestions for Authors
The aim of the work submitted for evaluation was to explore the synergy in both frequency and movement domain based on the non-negative tucker decomposition (NTD) method. The work is interestingly written, but in my opinion it requires supplementation in several places. Here are some detailed comments:
1. There is no clearly stated research hypothesis
2. The introduction and summary lack the practical application of the research conducted
3. The way of presenting data requires improvement. The Figures included in the work are not very legible, and especially Figure 3 is unclear, the lines in it are blurred and the colors are indistinct.
4. Legends should be placed under the tables.
5. Please explain the abbreviation "sEMG" in the Introduction (page 1, line 41), although it is explained in the abstract.
6. And please explain whether the authors notice any limitations of the conducted study.
Author Response
Dear reviewers: Thank you for your letter and for the reviewer’s comments concerning our manuscript entitled “Muscle synergy during wrist movements based on non-negative tucker decomposition” (ID: Sensors-2934544-2024). Those comments are all valuable and very helpful for revising and improving our paper, as well as the important guiding significance to our researches. We have studied those comments carefully and made corrections which we hope meet with approval. The main corrections are shown in red font in the revised manuscript. The respond to the reviewer’s comment is as follows:
(1). There is no clearly stated research hypothesis.
Answer: Thanks to your advice. In our study, our hypothesis is that each task has a specific synergy pattern with a shared synergy between two tasks. We have supplemented this in the Introduction Section.
(2). The introduction and summary lack the practical application of the research conducted.
Answer: Thanks for your advice. Our findings on muscle synergy will be of great significance to motor control and even to clinical assessment techniques. We have supplemented this in the Introduction and Conclusion Sections.
(3). The way of presenting data requires improvement. The Figures included in the work are not very legible, and especially Figure 3 is unclear, the lines in it are blurred and the colors are indistinct.
Answer: Thanks for your suggestions. The data results of this study are presented in the form of graphs and tables. We have improved the clarity of all the graphs and improved some of the details of the graphs such as the horizontal coordinates in this paper.
(4). Legends should be placed under the tables.
Response: Thanks to the reviewer's reminder. The legend of all the charts in this article has been placed below the chart.
(5). Please explain the abbreviation "sEMG" in the Introduction (page 1, line 41), although it is explained in the abstract.
Answer: Thanks. We have corrected it and made the following changes to the paper: the existing methods for muscle synergy are largely achieved by applying the matrix factorization to surface electromyography (sEMG) data.
(6). And please explain whether the authors notice any limitations of the conducted study.
Answer: Thanks for your suggestion. This study utilized wavelet decomposition and non-negative tensor decomposition to investigate the time-frequency-space characteristics of muscle coordination under different upper-limb movements. The exploration frequency band ranged from 0 to 150Hz, without further subdivision into frequency bands such as alpha, beta, and gamma bands. In addition, we only explored the multidomain muscle synergy features in healthy people, and less mentioned in patients with motor dysfunction, such as stroke. For patients, the activation and inhibition of multidomain muscle synergies may be relatively diminished, resulting in a joint response to motor control. Therefore, exploring multi-domain muscle synergies may provide a more theoretical basis for clinical evaluation applications. We have supplemented the limitations in Discussion section.
Reviewer 2 Report
Comments and Suggestions for Authors
The paper presents the multi-domain muscle synergy during the wrist movements study based on the non-negative tucker decomposition method. Presented results can help to explore the multi-domain synergistic characteristics of multi-channel surface EMG signals which can improve its processing.
Minor proofreading is required to deal with some issues like the following:
(lines 159-160)Because the time, frequency and channel of sEMG signals are non-negative[33]. The factor matrix obtained by NTD decomposition areis non-negative. (lines 210-211)According to the above analysis, we use NTD canto obtain the multidimensional characteristics of muscle synergy.(lines 211-212)It can not only decompose the frequency domain characteristics of muscle synergy. But also identify the shared muscle synergy and specific synergy between different movements (these divisions of one sentence on two are appears multiple times across the text).
One more thing that can be - but not necessary - considered as a flaw is that authors are comparing the results obtained with Nonnegative Tucker Decomposition (NTD) method with the results obtained with Nonnegative Tensor Factorization (NTF) method. The issue is that NTD is by definition the extension of NTF and better performance or additional functionality is expected (see [1]). It may be useful in terms of the paper's contribution clarity to add some comments about existing methods with similar functionality or to point out more specifically that there are no such methods considered in existing publications. [1] Y. -D. Kim and S. Choi, "Nonnegative Tucker Decomposition," 2007 IEEE Conference on Computer Vision and Pattern Recognition, Minneapolis, MN, USA, 2007, pp. 1-8, doi: 10.1109/CVPR.2007.383405. Comments on the Quality of English LanguageMinor proofreading is required.
Author Response
Dear reviewers: Thank you for your letter and for the reviewer’s comments concerning our manuscript entitled “Muscle synergy during wrist movements based on non-negative tucker decomposition” (ID: Sensors-2934544-2024). Those comments are all valuable and very helpful for revising and improving our paper, as well as the important guiding significance to our researches. We have studied those comments carefully and made corrections which we hope meet with approval. The main corrections are shown in red font in the revised manuscript. The respond to the reviewer’s comment is as follows:
(1). Minor proofreading is required.
Answer: Thanks for kindly advices. Based on the reviewers' suggestions, we have done grammar proofreading.
(2). One more thing that can be - but not necessary - considered as a flaw is that authors are comparing the results obtained with Nonnegative Tucker Decomposition (NTD) method with the results obtained with Nonnegative Tensor Factorization (NTF) method. The issue is that NTD is by definition the extension of NTF and better performance or additional functionality is expected (see [1]). It may be useful in terms of the paper's contribution clarity to add some comments about existing methods with similar functionality or to point out more specifically that there are no such methods considered in existing publications. [1] Y. -D. Kim and S. Choi, "Nonnegative Tucker Decomposition," 2007 IEEE Conference on Computer Vision and Pattern Recognition, Minneapolis, MN, USA, 2007, pp. 1-8, doi: 10.1109/CVPR.2007.383405.
Answer: Thanks to the reviewer's suggestion. In this study, a comparative analysis of muscle coordination level is carried out based on non-negative tensor decomposition and non-negative matrix factorization. Based on the reviewer's suggestions, we indicate the lack of studies on muscle synergy analysis based on NTD In the introduction section.
Reviewer 3 Report
Comments and Suggestions for Authors
Thank you for the opportunity to review the manuscript entitled „Muscle Synergy during Wrist Movements Based on Non-Negative Tucker Decomposition”. The aim of the study was to explore the synergy in both frequency and movement domain based on the non-negative tucker decomposition (NTD) method. Based on the obtained results authors concluded that t NTD is a meaningful method to explore the multi-domain synergistic characteristics of multi- channel sEMG signals.
The introductory part lacks justification for taking up the research topic, which should be improved. Some of the information included at the end of the introductory part is more relevant to the methodological part of this manuscript.
Is 10 tested people (7 males and 3 females) enough?
I am not convinced whether the description of the experiment can be described as an "Experimental Paradigm". It seems that "Paradigm" is a slightly broader concept. But that's probably just semantics.
The description of the methods is quite complicated and difficult to read.
The results could also be presented in a somewhat accessible description.
The discussion is a partial repetition of the results. There are few comparisons of the obtained results with other studies.
Does this work have any limitations of this research?
The conclusions from these studies lack a practical (application) conclusion.
Author Response
Dear reviewers: Thank you for your letter and for the reviewer’s comments concerning our manuscript entitled “Muscle synergy during wrist movements based on non-negative tucker decomposition” (ID: Sensors-2934544-2024). Those comments are all valuable and very helpful for revising and improving our paper, as well as the important guiding significance to our researches. We have studied those comments carefully and made corrections which we hope meet with approval. The main corrections are shown in red font in the revised manuscript. The respond to the reviewer’s comment is as follows:
(1). The introductory part lacks justification for taking up the research topic, which should be improved. Some of the information included at the end of the introductory part is more relevant to the methodological part of this manuscript.
Answer: Thanks. The topic selection basis has been explained and supplemented in the introduction section.
(2). Is 10 tested people (7 males and 3 females) enough?
Answer: Thanks. Considering that this study primarily investigates the time-frequency-space characteristics of muscle coordination, we refer to some relative existing literatures [1-3] to select the number of subjects, which consistently utilized 10 or fewer subjects. Additionally, ten subjects enrolled in this study also satisfies the requirements for statistical analysis. So we choose ten subjects in this study.
[1] J. Zariffa, J. Steeves, D.K. Pai, Changes in hand muscle synergies in subjects with spinal cord injury: characterization and functional implications, J. Spinal Cord Med. 35 (2012) 310-318.[2] N. Jiang, H. Rehbaum, I. Vujaklija, B. Graimann, D. Farina, Intuitive, online, simultaneous, and proportional myoelectric control over two degrees-of-freedom in upper limb amputees, IEEE Trans. Neural Syst. Rehabil. Eng. 22 (2014) 501-510.[3] F. Lunardini, C. Casellato, M. Bertucco, T.D. Sanger, A. Pedrocchi, Children with and without dystonia share common muscle synergies while performing writing tasks, Ann. Biomed. Eng. 45 (2017) 1949-1962.
(3). I am not convinced whether the description of the experiment can be described as an "Experimental Paradigm". It seems that "Paradigm" is a slightly broader concept. But that's probably just semantics.
Answer: Thanks for your advice. We have changed the term ‘experimental paradigm’ to ‘Experiments’.
(4). The description of the methods is quite complicated and difficult to read.
Answer: Thanks. This study primarily investigates the time-frequency-space characteristics of muscle coordination in different upper limb movements using non-negative tensor decomposition (NTD), and compares it with the non-negative matrix factorization (NMF). The NTD method includes the principle of non-negative matrix decomposition and the selection method for corresponding cooperative modules, which depends on the variability accounted for (VAF). Subsequently, the computational principle of non-negative tensor decomposition is presented, highlighting that the main difference between non-negative matrix decomposition and non-negative tensor decomposition lies in how to prepare the original data model for decomposition. Specifically, prior to conducting non-negative tensor decomposition in this study, it is essential to construct a third-order tensor time-frequency-space model. This involves transforming the original data into the time-frequency domain through wavelet decomposition and then constructing a third-order tensor model based on data from different channels. Finally, the third-order tensor model is decomposed using non-negative tensor decomposition. Similarly, an explanation of how to select the number of cooperative modules is provided, which depends on the fitting value (FIT) method.
(5). The results could also be presented in a somewhat accessible description.
Answer: Thanks for your suggestion. In this study, our aim is to explore the time-frequency-space characteristics of muscle coordination under different movements of upper limbs based on non-negative tensor decomposition, and our results show that non-negative tensor decomposition can also directly extract shared synergy and special synergy. To verify and compare the results, we compared these results with the non-negative matrix decomposition. In the result analysis part, firstly, based on the non-negative matrix decomposition, the spatial and temporal characteristics of muscle coordination are explored, and the muscle coordination module is extracted and analyzed. Then, the time-frequency-space characteristics of muscle coordination are analyzed based on the non-negative tensor decomposition. The synergistic similarity of muscles in different movements of the same subject and the same movements of different subjects were analyzed, and the inherent law of synergistic similarity of muscles was revealed. Finally, in order to verify the direct extraction characteristics of muscle synergy by non-negative tensor decomposition, Spearman correlation analysis was used to extract shared synergy and special synergy based on non-negative matrix decomposition, and then shared synergy and special synergy were extracted and compared based on non-negative tensor decomposition. The results show that the results of the two methods are consistent, which shows the efficiency and convenience of non-negative tensor decomposition. In this paper, the description of the result part has been improved and perfected.
(6). The discussion is a partial repetition of the results. There are few comparisons of the obtained results with other studies.
Answer: Thanks to the reviewer's suggestion. Comparisons with other studies are added in the discussion section of this paper.
For the feature analysis of muscle coordination, previous studies mainly extracted and analyzed the spatio-temporal features of muscle coordination based on non-negative matrix decomposition. Xiao L et al. introduced the non-negative matrix factorization (NMF) method to explore the muscle activation patterns and synergy structure under 6 types of movements. De Marchis C et al. analyzed the distribution of neural actuations among muscles that work together during the execution of a free pedal task based on nonnegative matrix factorization. However, these studies have neglected the frequency domain features of muscle coordination. Previous studies have shown that the motion control module has temporal and rhythmic characteristics. Xie P et al. proposed a novel method, named time-frequency non-negative matrix factorization (TF-NMF), to explore the time-varying regularity of muscle synergy characteristics of multi-channel surface electromyogram (sEMG) signals at different frequency bands. However, these studies show that the research based on NMF cannot analyze the multi-domain muscle synergy features. In this study, the potential application of higher-order tensor model in extracting multi-domain synergies were explored.
This study also analyzes the shared and specific characteristics of muscle synergy under different movement stages. Ahmed Ebied et al. showed that nonnegative tensor decomposition can be used for the direct extraction of shared muscle synergies and special muscle synergies by exploring the potential application of higher-order tensor models in myoelectric control[27]. According to the analysis, the ED, ECR, ECU and B muscles are specific synergy modules of WE, while the FCR, PL and FDS are specific synergy modules of WF. BB was shared muscle synergy module, which supports the wrist flexion and extension and maintains the stability of the upper limb [18], this conclusion is consistent with the results of studies on muscle activation and muscle synergism in different movement modes of the upper limb[40].
(7). Does this work have any limitations of this research?Answer: Thanks to the reviewer's question. This study utilized wavelet decomposition and non-negative tensor decomposition to investigate the time-frequency-space characteristics of muscle coordination. The exploration frequency band ranged from 0 to 150Hz, without further subdivision into frequency bands such as alpha, beta, and gamma bands. In addition, we only explored the multidomain muscle synergy features in healthy people, and less mentioned in patients with motor dysfunction, such as stroke. For patients, the activation and inhibition of multidomain muscle synergies may be relatively diminished, resulting in a joint response to motor control. Therefore, exploring multi-domain muscle synergies provides a more theoretical basis for clinical evaluation applications. The limitations have been supplemented in the Discussion section.
(8). The conclusions from these studies lack a practical (application) conclusion.
Answer: Thanks to the reviewer's reminder. Our findings on muscle synergy will be of great significance to motor control and even to clinical assessment techniques. Changes have been made in the article.
Reviewer 4 Report
Comments and Suggestions for Authors
Authors investigated explore the synergy in both frequency and movement domain based on the non-negative tucker decomposition using surface electromyography. Overall Figure quality looks not good. In addition, description of the results are further required to be explained in detail.
1) Abstract format is wrong. Authors need to check author guidelines.
2) If the Figure 1 is obtained from another reference, please provide the copyright permission.
3) Authors need to check entire Figure quality because the image quality is not so good.
4) Authors need to use the names of the abbreivated journal in Reference.
5) Author Contributions format is also wrong.
6) Authors must specify how to choose multi-point of surface electromyography.
7) Authors must use formal English expression in the manuscript such as and.
8) In Table 3, correlation coefficient values seems to be a little bit high. Is there any reason ?
9) Authors compared WF NMF in Figure 4 and NTD in Figure 5. However. it is hard to understand why proposed method showed better results so authors had better describe the advantages and disadvantages.
10) In Tables 1 and 2, synergistic muscles of all subjects are shown such as PL FDS BB, BB.. What represent them ?
11) In Figure 6, there are no units in x-axis.
Comments on the Quality of English Language
Englsih grammars need to be checked by native English colleagues because of the insufficient description.
Author Response
Dear reviewers: Thank you for your letter and for the reviewer’s comments concerning our manuscript entitled “Muscle synergy during wrist movements based on non-negative tucker decomposition” (ID: Sensors-2934544-2024). Those comments are all valuable and very helpful for revising and improving our paper, as well as the important guiding significance to our researches. We have studied those comments carefully and made corrections which we hope meet with approval. The main corrections are shown in red font in the revised manuscript. The respond to the reviewer’s comment is as follows:
(1). Abstract format is wrong. Authors need to check author guidelines.
Answer: Thanks. According to the authors’ guidelines, we have corrected and adjusted the layouts of this Abstract section in the new manuscript.
(2). If the Figure 1 is obtained from another reference, please provide the copyright permission.
Answer: Thanks for your advice. Figure 1 is our original drawing.
(3). Authors need to check entire Figure quality because the image quality is not so good.
Answer: Thanks for your advice. We have improved all figures to make them clearer in the new manuscript.
(4). Authors need to use the names of the abbreviated journal in Reference.
Answer: Thanks. We have adjusted he names of the abbreviated journal of all literatures cited in the new manuscript.
(5). Author Contributions format is also wrong.
Answer: Thank you. Author Contributions has been modified according to the correct format.
(6). Authors must specify how to choose multi-point of surface electromyography.
Answer: Thanks. Based on the anatomical and physiological knowledge of the functional upper limb and specific movements, the corresponding muscle position can be identified, and the electrodes were placed in the muscle belly position along the direction of the muscle fibers.
(7). Authors must use formal English expression in the manuscript such as and.
Answer: Thanks. English grammar has been corrected according to the reviewer's suggestion.
(8). In Table 3, correlation coefficient values seem to be a little bit high. Is there any reason?
Answer: Thanks to the reviewer's question. Similarity between two muscle synergies matrices were estimated by Pearson's correlation coefficients ranging from 0 to 1, and large value means high similarity. When , they are considered similar [1]. The high correlation coefficient proves that the corresponding modules share high coordination in the process of wrist flexion and wrist extension.
[1] Chen X, et al., Muscle activation patterns and muscle synergies reflect different modes of coordination during upper extremity movement. Front Hum Neurosci. 2023. 16: 912440.
(9). Authors compared WF NMF in Figure 4 and NTD in Figure 5. However. it is hard to understand why proposed method showed better results so authors had better describe the advantages and disadvantages.
Answer: Non-negative matrix decomposition can only explore the spatio-temporal characteristics of muscle coordination, but the calculation is relatively simple, while non-negative tensor decomposition can explore the time-frequency-space characteristics of muscle coordination, and can directly extract shared cooperation and special cooperation modules. However, tensor models need to be built before tensor decomposition, and the calculation is relatively complex.
(10). In Tables 1 and 2, synergistic muscles of all subjects are shown such as PL FDS BB, BB. What represent them?
Answer: Thanks. They represent the synergistic muscles in each module of each subject. 8 muscles of the right upper limb include the biceps brachii (BB), brachioradialis (B), flexor carpi radialis (FCR), palmaris longus (PL), extensor carpi radialis (ECR), extensor digitorum (ED), extensor carpi ulnaris (ECU), and flexor digitorum superficialis (FDS). We introduced it in the experimental section
(11). In Figure 6, there are no units in x-axis.
Answer: Thanks. We have added the units in x-axis in Figure 6.
Reviewer 5 Report
Comments and Suggestions for Authors
The study by Chen and colleagues aims to assess muscle synergies in the time-frequency-space domain, involving tensor decomposition through NTD (Nonnegative Tensor Decomposition). Only a few pieces of literature about the use of tensor decomposition for synergy extraction have been found in neuromuscular control, and the idea may hold some potential. Indeed, classical NMF (Nonnegative Matrix Factorization) considers only the time and space domains, and although it is a consolidated and useful approach, it has limitations. However, at this stage, the paper does not demonstrate a consistent advantage of passing through wavelet transform and NTD compared to NMF. Some points need clarification to better frame the potential of the study and understand what additional information one can obtain using the proposed methods. Below, I have outlined major and minor points that should be addressed.
Major points:
1) All the figures in the paper shold be improved in terms of resolution. It is difficult to read figures especially the one in the results. This does not allow a clear evaluation of the manuscript from the readers.
2)In the Methods section, Authros lack important citations related to typical NMF, in particular the criterion for identifying the optimal number of synergies should be at list reported. For this reason I suggest the Authros to report the following papers:
- "Comparison of initialization techniques for the accurate extraction of muscle synergies from myoelectric signals via nonnegative matrix factorization." Applied bionics and biomechanics 2018 (2018).
-An objective, information-based approach for selecting the number of muscle synergies to be extracted via non-negative matrix factorization." IEEE Transactions on Neural Systems and Rehabilitation Engineering 29 (2021): 2676-2683.
3)Plaese specify if the range of frequency are the results of wavlet filter bank decomposition.
4) I think that it is important not only compare correlation of synergies between WE and WF but also quantify how much the set of synergies hold intersubject. Indeed one may wonder if for a certain range of frequency the synergies are correlated among subjects
Minor points:
1) In the Introduction Authros mainly introduced more the application of synergies in terms of analysis. However, it can be beneficial to stress also that synergies can be applied in modern myoelectric control for rehabilitation technologies. Thus I suggest the Authros to review the following studies:
-"On the Decoding of Shoulder Joint Intent of Motion from Transient EMG: Feature Evaluation and Classification." IEEE Transactions on Medical Robotics and Bionics (2023).
-"Handwritten digits recognition from sEMG: Electrodes location and feature selection." IEEE Access (2023).
2)It is important to do tests and discuss about the ability of NTD to generalize the standard NMF.
Author Response
Dear reviewers: Thank you for your letter and for the reviewer’s comments concerning our manuscript entitled “Muscle synergy during wrist movements based on non-negative tucker decomposition” (ID: Sensors-2934544-2024). Those comments are all valuable and very helpful for revising and improving our paper, as well as the important guiding significance to our researches. We have studied those comments carefully and made corrections which we hope meet with approval. The main corrections are shown in red font in the revised manuscript. The respond to the reviewer’s comment is as follows:
(1). All the figures in the paper should be improved in terms of resolution. It is difficult to read figures especially the one in the results. This does not allow a clear evaluation of the manuscript from the readers.
Answer: Thanks for your advice. This issue has been corrected, and we have made changes in the new manuscript.
(2). In the Methods section, Authors lack important citations related to typical NMF, in particular the criterion for identifying the optimal number of synergies should be at list reported. For this reason I suggest the Authors to report the following papers:- "Comparison of initialization techniques for the accurate extraction of muscle synergies from myoelectric signals via nonnegative matrix factorization." Applied bionics and biomechanics 2018 (2018).-An objective, information-based approach for selecting the number of muscle synergies to be extracted via non-negative matrix factorization." IEEE Transactions on Neural Systems and Rehabilitation Engineering 29 (2021): 2676-2683.
Answer: Thanks. Recommended references are cited and presented in the article: For the determination of the number of muscle synergies, Ranaldi S et al [33] proposed an Akaike Information Criterion (AIC)-based method for model order selection when extracting muscle synergies via the original Gaussian Non-Negative Matrix Factorization algorithm, Soomro MH et al [34] assessed the performance of different initializations of matrix factorization algorithms for an accurate identification of muscle synergies.
(3). Please specify if the range of frequency are the results of wavelet filter bank decomposition.
Answer: Thanks to the reviewer’s question. The range of frequency are the results of wavelet filter bank decomposition. It is described in 2.3.2 section.
(4). I think that it is important not only compare correlation of synergies between WE and WF but also quantify how much the set of synergies hold intersubject. Indeed one may wonder if for a certain range of frequency the synergies are correlated among subjects.
Answer: Thanks to the reviewer’s question. This paper mainly analyzes the characteristics of muscle synergy based on non-negative tensor decomposition and non-negative matrix decomposition. From the comparison of the results, it can be seen that non-negative tensor decomposition can be used to explore the time-frequency-space characteristics of muscle synergy, and can directly extract the shared muscle synergy and special muscle synergy components under different movements. The frequency domain characteristics of muscle synergy can be seen from the NTD decomposition results. In the part of muscle synergistic similarity analysis, this paper carried out muscle synergistic similarity analysis under the same movement of different subjects and different movements of the same subject respectively, and the results showed that the muscle synergistic similarity under the same movement of different subjects was higher. The muscle synergistic similarity of different subjects in specific frequency domain needs to be further studied and analyzed.
(5). In the Introduction Authors mainly introduced more the application of synergies in terms of analysis. However, it can be beneficial to stress also that synergies can be applied in modern myoelectric control for rehabilitation technologies. Thus I suggest the Authors to review the following studies:-"On the Decoding of Shoulder Joint Intent of Motion from Transient EMG: Feature Evaluation and Classification." IEEE Transactions on Medical Robotics and Bionics (2023).-"Handwritten digits recognition from sEMG: Electrodes location and feature selection." IEEE Access (2023).
Answer: Thanks for your suggestion. According to the reviewer's suggestion, the recommended literature has been cited in this study. Previous studies have shown that the study of muscle synergy characteristics is more helpful to understand the way that CNS controls the production, execution and coordination of human movement [4], which can also be applied in modern myoelectric control for rehabilitation technologies [5, 6].
(6). It is important to do tests and discuss about the ability of NTD to generalize the standard NMF.
Answer: Thanks to the reviewer’s suggestion. In the introduction, the relationship between NTD and NMF is discussed. Moreover, it can be seen from the analysis of the decomposition results in this study that NTD has a good generalization ability to NMF and can be used for the decomposition of three-dimensional data. The decomposition results reflect the time-frequency-space multi-dimensional intrinsic correlation characteristics of muscle coordination.
Round 2
Reviewer 3 Report
Comments and Suggestions for Authors
Thank you for the opportunity to review this manuscript again. After re-reading and the authors' responses to the review, I believe that the manuscript is suitable for publication in the journal.
Reviewer 5 Report
Comments and Suggestions for Authors
Authors addressed all my concerns in the revised version of the maanuscript.